# Reformulating Zero-shot Action Recognition for Multi-label Actions

**Alec Kerrigan**     **Kevin Duarte**     **Yogesh S Rawat**     **Mubarak Shah**

Center for Research in Computer Vision, University of Central Florida, Orlando, FL 32816,
{aleckerrigan,kevin_duarte}@knights.ucf.edu, {yogesh,shah}@crcv.ucf.edu

## Abstract

The goal of zero-shot action recognition (ZSAR) is to classify action classes which were not previously seen during training. Traditionally, this is achieved by training a network to map, or regress, visual inputs to a semantic space where a nearest neighbor classifier is used to select the closest target class. We argue that this approach is sub-optimal due to the use of nearest neighbor on static semantic space and is ineffective when faced with multi-label videos - where two semantically distinct co-occurring action categories cannot be predicted with high confidence. To overcome these limitations, we propose a ZSAR framework which does not rely on nearest neighbor classification, but rather consists of a pairwise scoring function. Given a video and a set of action classes, our method predicts a set of confidence scores for each class independently. This allows for the prediction of several semantically distinct classes within one video input. Our evaluations show that our method not only achieves strong performance on three single-label action classification datasets (UCF-101, HMDB, and RareAct), but also outperforms previous ZSAR approaches on a challenging multi-label dataset (AVA) and a real-world surprise activity detection dataset (MEVA).

## 1   Introduction

Current image and video classification models require large labeled training datasets but they perform really well on previously seen classes. However, if novel classes are presented to these models, as is the case in many real-world applications, they tend to fail. The system has to be retrained with additional samples for this class to correctly predict these novel classes, which requires additional training time as well as computational resources. Therefore, it would be beneficial if we can train a single system on a fixed dataset which can be applied to new, previously unseen classes. To this end, zero-shot learning (ZSL) approaches [1] have been proposed which leverage semantic information, e.g. textual descriptions or class names, and relate them to new unseen class categories.

In this work, we focus on zero-shot action recognition (ZSAR), where the goal is to classify videos of unseen action categories. There has been a great progress focusing on ZSAR [2, 3, 4], however, most of these existing methods utilize a similar fundamental approach. These approaches project a video representation to a fixed semantic space (e.g. a text embedding space generated from a pre-trained Word2Vec [5] model) and perform classification using a nearest neighbor operation. We argue that such a solution is sub-optimal because the class selection is performed on a static text-based semantic space. This formulation forces classes that are more similar in the semantic space to have closer classification boundaries, even if they are visually dissimilar; conversely, actions that are visually similar, can appear further apart in the semantic space. For example, the action "Tennis Swing" is closer in semantic space to "Swing" (i.e. a child on a swing) than "Table Tennis Shot" even though it is more visually similar to the latter action.

35th Conference on Neural Information Processing Systems (NeurIPS 2021).

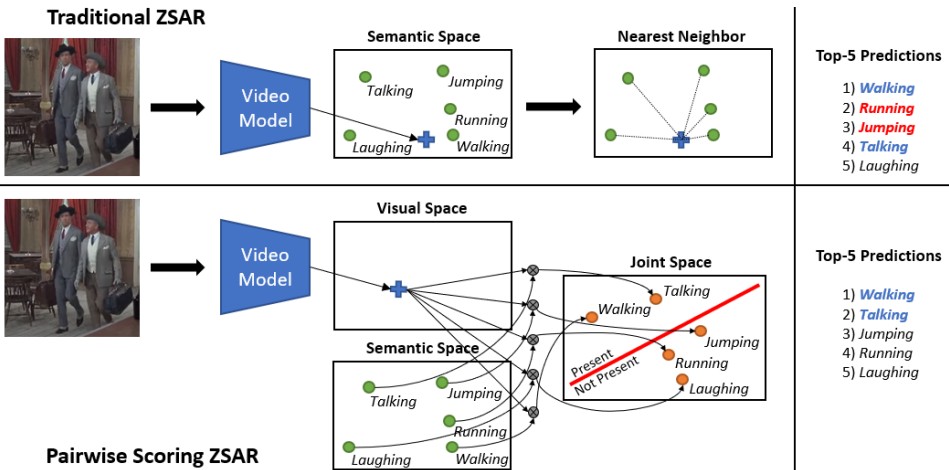

Figure 1: C onventional nearest neighbor ZSAR methods (top) cannot predict semantically dissimilar actions "Walking" and "Talking" without actions "Running" and "Jumping" also being predicted. Our proposed pairwise scoring approach (bottom), can predict two dissimilar classes by mapping the video and semantic embeddings to a joint embedding space which can be linearly separated.

In addition, this is detrimental in the multi-label case where multiple semantically distinct actions can occur within a sample. This is illustrated in Figure 1 (top). In conventional ZSAR approaches, the video network can only generate a single representation in the semantic space so the actions "Walking" and "Talking" can not both be predicted without many incorrect action categories also being predicted. This issue could be solved by fine-tuning the textual encoder which generates the semantic space (i.e. moving the representations of "Walking" and "Talking" closer together). However, previous approaches train the video models by regressing to the action class's semantic feature vector. If the vector is not fixed (i.e. the text encoder is fine-tuned), this would collapse to a trivial solution: it can map all classes to zero vectors which are easily regressed to by the video model. Several constraints could prevent this collapse, however such solutions to the best of our knowledge, have not been proposed in the ZSAR literature.

We propose a novel solution to this problem where a model finds agreement between the features from a pair of inputs (i.e. the video and a semantic/textual feature vector) and outputs a match vs. no-match probability score. Our method consists of a pairwise scoring function which has two main benefits. First, the model can refine the input text features through additional parameterized layers, without collapsing to a trivial solution. Second, the method outputs individual action probability scores used for classification instead of relying on nearest neighbor. Figure 1 (bottom) depicts this behaviour. The Pairwise Scoring ZSAR (PS-ZSAR) approach is able to map the merged visual and semantic representations into a joint space, on which a decision boundary can be created to successfully predict both "Walking" and "Talking" without predicting irrelevant actions.

PS-ZSAR is trained end-to-end using standard classification losses. We evaluate our proposed network on five video datasets. On three single-label action recognition datasets - UCF-101, HMDB, and RareAct - it achieves strong performance when compared to previous approaches. We report, to the best of our knowledge, the first results for zero-shot multi-label action recognition on the AVA dataset, which we believe will be a challenging baseline for future ZSAR methods. Lastly, we use PS-ZSAR in the NIST ActEV challenge "Surprise Activity" task which demonstrates the method's ability to scale to difficult real-world scenarios.

## 2  Related Work

There is a vast literature pertaining the zero-shot action recognition (ZSAR) [1]. There are bodies of work on *transductive* ZSAR [6, 7, 8, 9], where test data (i.e. videos) are available during training but classes are not, and *generalized* ZSAR (GZSAR) [10, 11, 12, 13, 14, 15] which evaluates method performance on both seen and unseen classes. However, in this work, we concentrate on *inductive*

ZSAR in which the test data and classes are unknown during training and the method is evaluated only on unseen classes.

## 2.1 Zero-shot Action Recognition

Typical zero-shot video classification systems [16, 17, 3, 18, 19] extract visual features from frames or 3D video clips using pre-trained image or video networks, resepctively (e.g. ResNet [20] or C3D [21] models), and then train a separate model that maps the visual features to a semantic embedding space. For ZSAR methods, if video networks generalize well to the semantic space, the model can perform well on semantic embeddings for classes previously unseen during training. At test time, these methods predict the class which has semantic embedding that is the nearest neighbor of the model's output. The semantic space usually takes the form of text embeddings from a Word2Vec [5] model. In other methods [22, 23, 24], the semantic space consists of attributes manually selected by an expert; however, such methods are difficult to apply to real-world scenarios.

There have been a variety of methods to learn models which can map video representations to semantic space. Zhu *et al.* [2] treat ZSAR as a generalized multi-instance learning problem and learn a kernelized representation that can be directly compared with unseen action prototypes. Action2Vec [3] uses a combination of supervised classification loss and a ranking loss to map video features to textual space. Recently, Brattoli *et al.* [4] showed that large performance increases can be obtained by learning the video model rather than using pre-extracted video features. We find that this idea can be extended to the text model: by learning improved text representations, i.e. refining the static Word2Vec embeddings, we achieve improved generalization on unseen classes.

## 2.2 Multi-label Zero Shot Learning

Multi-label zero shot learning has been explored in limited capacity, primarily for images. Lee [25] utilized a graph structure to learn semantic dependencies between words in a class name to learn relevant textual features. Huynh and Elhamifar [26] used an attention framework to map important segments of an image to a join visual-label embedding space for classification. Huang [27] used a similar approach while also proposing transferring learned embedding spaces from systems trained on ImageNet. However, multi-label zero shot learning for action recognition is a far less explored space. To the best of our knowledge, Wang and Chen [28] propose the first work to address this problem. Their work focuses on learning temporal relationships between frames in a video, then using a ranking loss to learn alongside a semantic encoder. In this work, we simplify the multi-label ZSAR task by proposing a pairwise scoring network that can be trained end-to-end with cross-entropy loss. Furthermore, we evaluate on two multi-label datasets, AVA and MEVA, and show that our approach can scale to these challenging benchmarks.

# 3 Method

## 3.1 Problem Formulation

Zero-shot action recognition aims to classify unseen actions, $\mathcal{U}$, by transferring knowledge learned from seen training classes, $\mathcal{S}$. The training set, $D_S = \{(x, y) | x \in \mathcal{X}, y \in \mathcal{S}\}$, consists of labeled videos $x$ in video space $\mathcal{X}$ and label $y$. The test set is defined as $D_U = \{(x, y) | x \in \mathcal{X}, y \in \mathcal{U}\}$. The action classes in both datasets must be disjoint, i.e. $\mathcal{S} \cap \mathcal{U} = \emptyset$. The goal of ZSAR is to predict a label for a videos of unseen class (activities), i.e. $\mathcal{X} \to \mathcal{U}$.

For each class, a semantic embedding vector $\psi(y) \in \mathbb{R}^D$ of length $D$ is obtained. These vectors are traditionally obtained by averaging word2vec [5] embeddings for each word in the class name, however different text encoders (e.g. sent2vec [29], BERT [30]) could also be used. Conventional ZSAR methods learn a parameterized model $f_\theta$ which maps an input video to the semantic space, i.e. $f_\theta : \mathcal{X} \to \mathbb{R}^D$. Then, for a given video, $x$, a classification is performed by a nearest neighbor in the set of test class embeddings, as defined by:

$$F(x) = \underset{y \in \mathcal{U}}{\arg\min}\, d(f_\theta(x), \psi(y)), \tag{1}$$

where $d$ is a distance metric (e.g. cosine distance).

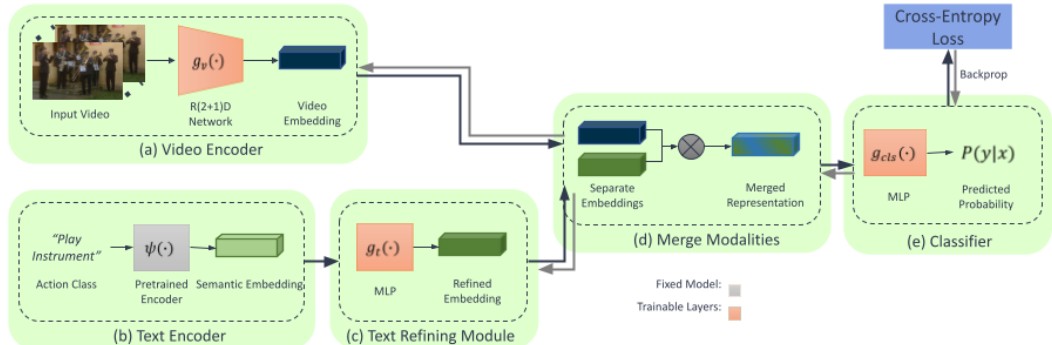

Figure 2: **Overview of our proposed network architecture.** Our method consists of four main parts: (a) The Video Encoder which extracts video features from a given video clip, (b) the Text Encoder which generates the semantic embedding from the action class name or natural language description, (c) Text Refining module which fine-tunes the semantic embeddings during training, and (d) the Scoring Function which generates the probability that the pair of inputs match. The network is trained end-to-end using cross-entropy loss.

We argue that this two stage approach - metric learning on the semantic space and nearest neighbor classification - is ineffective. Instead, we reformulate the ZSAR problem to learn a scoring function $h_\theta$ which predicts a match vs. no-match probability when given a video and semantic embedding pair, i.e. $h_\theta : \mathcal{X} \times \mathbb{R}^D \to [0, 1]$. Once the network is trained, the final action classification can be obtained by selecting the class which achieves the highest probability:

$$F(x) = \underset{y \in \mathcal{U}}{\operatorname{argmax}}\, h_\theta\left(x, \psi\left(y\right)\right). \tag{2}$$

This formulation allows for multi-label action recognition by selecting all classes with probabilities over the threshold $\tau$, i.e.

$$F(x) = \{y \in \mathcal{U} | h_\theta\left(x, \psi\left(y\right)\right) > \tau\}, \tag{3}$$

whereas nearest neighbor classification can not predict multiple semantically dissimilar actions to occur within a given video.

### 3.2 Proposed ZSAR Framework

Our proposed network architecture is illustrated in Figure 2. Here, we describe the different components of our architecture.

**Video and Text Encoders**   Given a video clip $x \in \mathbb{R}^{T \times H \times W}$ of $T$ frames with a height and width of $H$ and $T$ respectively, we learn a 3D-CNN [21, 31], $g_v$, to extract a video feature representation, $g_v\left(x\right) \in \mathbb{R}^{D_v}$. Also, we obtain the semantic embedding (text encoding), $\psi\left(y\right)$, by passing the action class name through a fixed pre-trained text encoder. Unless otherwise stated, a Word2Vec embedding is created for each word and the embeddings over all words in the action name are averaged to obtain a 300 dimensional text feature representation, $\psi\left(y\right) \in \mathbb{R}^{300}$. We also present results with the more recent Sent2Vec text encoder in Section 4.4.

**Refining Text Embedding**   Since text encoders are trained on large text corpora, they tend to lack information about visual attributes of actions. For example, "running water" and "person running" are visually distinct, but embedding for the word "running" is the same for both. To remedy this, we propose a module, $g_t$, which refines, or fine-tunes, the text encodings to learn vision-specific representations and generates a new text representation with dimension $D_t$: $g_t\left(\psi\left(y\right)\right) \in \mathbb{R}^{D_t}$.

**Merging Features and Scoring**   With the extracted visual features and refined textual features, we need to determine whether or not the inputs match. We first merge the features to obtain a single representation for the pair of inputs, $m\left(g_v\left(x\right), g_t\left(\psi\left(y\right)\right)\right) \in \mathbb{R}^{D_m}$. This merging operation can take many forms (concatenation, addition, multiplication, etc.), which are explored in Section 4.4.

Once merged, the joint feature representation is passed through an MLP, $g_{\text{cls}}$, to obtain classification logits:

$$f_{\text{logits}}\left(x, \psi\left(y\right)\right) = g_{\text{cls}}\left(m\left(g_v\left(x\right), g_t\left(\psi\left(y\right)\right)\right)\right). \tag{4}$$

Then, final probability score is obtained by using a softmax operation across all classes:

$$P\left(y|x\right) = h_\theta\left(x, \psi\left(y\right)\right) = \frac{\exp\left(f_{\text{logits}}\left(x, \psi\left(y\right)\right)\right)}{\sum\limits_{y' \in \mathcal{S}} \exp\left(f_{\text{logits}}\left(x, \psi\left(y'\right)\right)\right)}. \tag{5}$$

The final probability scores are then be used for either single-label or multi-label classification following equations 2 and 3, respectively.

**Training Procedure**   We train the network in an end-to-end manner using cross-entropy loss. When training on a single-label dataset, we compute the loss

$$\mathcal{L} = -\frac{1}{B} \sum_{i=1}^{B} \log P\left(y_i|x_i\right), \tag{6}$$

on a mini-batch of samples $B = \{\left(x_1, y_1\right), ..., \left(x_K, y_K\right)\}$. When training on multi-label data, which is the case for our experiments on the MEVA dataset, the softmax activation is replaced with sigmoid so that each probability is independent and binary-cross entropy loss is used.

## 4   Experimental Evaluation

**Implementation Details**   For our experiments, the video encoder is the PyTorch [32] implementation of the R(2+1)D-18 [31] network. This network outputs a visual embedding dimension of $D_v = 512$ for each 16-frame video clip. As with previous, work [4, 33] we average predictions over 25 clips per video at test time. Unless stated otherwise, the text encoder is a pretrained Word2Vec [5] model with max-pooling over words, resulting in a 300-dimensional text embeddings. We also present results using the Sent2Vec [29] model, which is pre-trained on twitter unigrams and results in a 700-dimensional embedding for a given sequence of words. Our Text Refining module consists of a learned 3-layer MLP with hidden dimensions of 1024 and ReLU activations, and a final output dimension of $D_t = 600$. The output is passed through a sigmoid activation to obtain an embedding in the range $[0, 1]$. The loss of our model is minimized using the Adam optimizer [34] with a starting learning rate of 1e-3 and a batch size of 114. The model is trained for 50 epochs with a learning rate decrease by a factor of 10 at epoch 30. All experiments are performed on two of Nvidia Tesla V100 GPUs.

**Training Dataset**   We train our models on the Kinetics 700 [35] action recognition dataset. Following [4], we ensure that these experiments remain true zero-shot tasks by filtering out classes from Kinetics that are not sufficiently different than our testing classes. This results in 545,317 training videos consisting of 662 action classes[1].

### 4.1   Single-label Action Recognition

**Evaluation Datasets**   We evaluate on the UCF-101 [36], HMDB [37], and RareAct [38] dataests. UCF101 has 101 action classes across 13320 videos taken from YouTube. HMDB has 51 human focused actions based around sports and everyday activities across 6767 videos sourced from commercial areas as well as YouTube. RareAct is a video dataset containing unusual actions such as "blend phone", "cut keyboard", and "microwave shoes". It contains 905 videos with 122 different actions created by combining rarely co-occuring action verb pairs.

**Evaluation Protocols**   To compare with previous ZSAR methods, we use two evaluation protocols on UCF-101 and HMDB. The first, proposed in [4], involves testing on all videos from each dataset. The second involves randomly selecting and evaluating on half of the actions from a dataset (50

---

[1]It should be noted that [4] trains on 664 classes, but due to differences in dataset versions (i.e. changed action names) we have 2 fewer classes for training.

Table 1: Evaluation on all classes of UCF-101 and HMDB datasets.

| Method | UCF-101 | | HMDB | | RareAct | |
|---|---|---|---|---|---|---|
| | Top-1 | Top-5 | Top-1 | Top-5 | Top-1 | Top-5 |
| URL[2] | 34.2 | - | - | - | - | - |
| Brattoli *et al.* [4] | 37.6 | 62.5 | 26.9 | 49.8 | 9.4 | 24.8 |
| PS-ZSAR (ours) | **40.1** | **66.3** | **27.3** | **55.7** | **11.5** | **32.8** |

Table 2: Evaluation on UCF-101 and HMDB-51 datasets. Accuracies are the average of 10 random splits each with 50% of the classes.

| | UCF101 | HMDB |
|---|---|---|
| DataAug [2] | 18.3 | 19.7 |
| InfDem [39] | 17.8 | 21.3 |
| BiDrectional [9] | 21.4 | 18.9 |
| FairZSL [40] | - | 23.1 |
| TARN [16] | 19.0 | 23.5 |
| Action2Vec [3] | 22.1 | 23.5 |
| Brattoli *et al.*[4] | 48.0 | 32.7 |
| PS-ZSAR (ours) | 49.2 | 33.8 |

for UCF-101 and 25 for HMDB). This is repeated 10 times and the averaged results are reported[2]. Since RareAct is composed of "positive", "negative", and "hard negative" samples, we consider only positive clips; we follow the first evaluation protocol and evaluate on all 1765 positive video samples.

**Comparison with State-of-the-art Methods**    For the first evaluation protocol, we compare to Zhu *et al.* [2] and Bratolli *et al.* [4]. The results are presented in Table 1. Our approach outperforms the previous state-of-the-art methods on both datasets, with a large improvement in Top-5 accuracy (3.8% and 5.9% improvements on UCF-101 and HMDB, respectively). We also observe this performance improvement when compared to other previous ZSAR methods using the second evaluation protocol in Table 2. The RareAct dataset presents a challenging benchmark for zero-shot action classification, because the actions are drastically different than those used during training. Again, we find that our approach outperforms the strongest baseline with a 2.1% improvement on Top-1 accuracy and 8% improvement in Top-5 accuracy.

### 4.2    Multi-label Action Recognition on AVA

**Evaluation Dataset**    The Atomic Visual Actions (AVA) dataset [41] annotates 80 atomic visual actions in 340 15-minute video clips. All people in each video are exhaustively annotated at a rate of one frame-per-second, and each person can perform multiple actions at once. We present results on the validation set which contains 64 videos split into 54k one-second clips.

**Evaluation Protocol**    Since action localization (i.e. predicting bounding-box localizations for all actors in a video) is out of scope for this work, we evaluate only our the action recognition capabilities on the AVA dataset. To this end, we extract video clips centered on each actor's ground-truth bounding-box, and pass these through the ZSAR model to obtain class predictions. For fair comparison with [4], we generate multiple predictions using several strategies which are outlined in the Supplemental Material. Ultimately, selecting all classes using a threshold on the distance from predictions to the text embedding leads to best scores, which are reported in this work. The resulting predictions are then used to compute the f-mAP metric as well as the standard multi-label classification metric F1-score.

As previous methods were not designed with multi-label action recognition, these systems must be modified to allow for comparison; specifically, to compute the mAP metric they must predict

---

[2]This evaluation protocol is greatly influenced by the random seed used to generate the splits. We discuss this further in the Supplementary Materials.

Table 3: Multi-label action recognition evaluation on the AVA Dataset. w2v indicates Word2Vec is used as the text encoder; s2v indicates Sent2Vec is used.

|  | Text Enc. | mAP | F1-score |
|---|---|---|---|
| Brattoli *et al.* [4] | w2v | 6.4 | 10.0 |
| PS-ZSAR (ours) | w2v | 6.5 | 11.4 |
| PS-ZSAR (ours) | s2v | **7.0** | **12.3** |
| Supervised (I3D) [44] | - | 23.4 | - |

Table 4: Results from the MEVA Sequestered Data Leaderboard. For both metrics, lower score is better.

|  | $P_{miss@0.02}T_{FA}$ | $P_{miss@0.04}T_{FA}$ |
|---|---|---|
| IBM-Purdue | 0.8898 | 0.8176 |
| CMU-DIVA | 0.8795 | 0.8044 |
| Purdue | 0.8537 | 0.7617 |
| UMCMU | 0.8724 | 0.8009 |
| PS-ZSAR | **0.7752** | **0.6686** |

confidence scores for their final classifications. We apply the method in [4] and obtain a confidence score for each class by computing the softmax over the inverse cosine distances between the video encoding and the classes' textual encoding. As most resultant distances are relatively close, we scale them by a factor of 10 prior to use in the softmax. For final class predictions, we follow equation 3, which we find leads to the best results for the baseline[3].

**Results**  We present results on the AVA dataset in Table 3. We find that, when using the same Word2Vec text encoder, our model outperforms previous approaches when modified for multi-label action classification in terms of F1-score. This improvement is more noticeable when using Sent2Vec as the text encoder - we achieve a 0.6% improvement in terms of mAP and a 2.3% improvement in F1-score.

### 4.3  Real-world Action Recognition on MEVA

**Dataset**  So far, the previous evaluation datasets have been curated to contain specific action classes. Also, unseen classes are identified by class names, rather than natural language sentences, which is not ideal for real-world applications. To demonstrate our approach's ability to deal with real-world scenario, we train and evaluate on the Multiview Extended Video with Activities (MEVA) dataset [42]. This dataset consists 9,300 hours of untrimmed video collected from multiple viewpoints and scenes. For 144 hours of video, bounding-box annotations are available for actors performing 37 possible activities. Each actor can perform multiple actions at once and *each activity has detailed natural language descriptions*, which is beneficial for learning ZSAR models. The data is split into 22 hours for training and 122 hours are sequestered for the NIST Activity in Extended Video (ActEV) challenge. This challenge has a "Surprise Activity" task, which involves classifying previously unseen classes (i.e. classes not within the annotated 37 activities) from natural language descriptions. As the test data is sequestered, *the number of unseen action categories as well as the number of instances are unknown.*

**Training and Evaluation Protocol**  On this dataset, PS-ZSAR is trained using the natural language descriptions of the activities (rather than their names). Since these descriptions consist of full-length sentences, we use Sent2Vec as our text encoder. Evaluation is performed by submitting a system to the ActEV challenge website. Performance is evaluated based on the probability of missed detection at fixed time-based false alarm per minute ($P_{miss}@T_{FA}$). The time-based false alarm per minute is set to 0.02 and 0.04 (i.e. $P_{miss@0.02}T_{FA}$ and $P_{miss@0.04}T_{FA}$). We refer to MEVA SDL [43] for detailed explanations of the evaluations as well as evaluation code.

**Results**  Our method achieves strong performance when compared with other submitted systems, as seen in Table 4. We find that our approach greatly outperform all other systems on the $P_{miss}@T_{FA}$ metric. These experiments highlight our method's ability to scale to large amounts of real-world video data as well as its ability to leverage natural language activity descriptions for the ZSAR task.

---

[3]We include various methods for obtaining confidence scores and final class predictions in the Supplementary Materials

Table 5: Ablation on Text Refining Module.

| Dataset | UCF-101 | | HMDB | |
|---|---|---|---|---|
| | Top-1 | Top-5 | Top-1 | Top-5 |
| No Refining | 32.0 | 58.5 | 18.9 | 45.6 |
| Single layer | 36.3 | 64.2 | 23.7 | 50.1 |
| Full Model | **40.1** | **66.3** | **27.3** | **55.7** |

Table 6: Ablation on how modalities are merged.

| Dataset | UCF-101 | | HMDB | |
|---|---|---|---|---|
| | Top-1 | Top-5 | Top-1 | Top-5 |
| Concat. | 34.4 | 60.6 | 21.3 | 47.7 |
| Add. | 35.1 | 62.6 | 24.0 | 51.6 |
| Mult. | **40.1** | **66.3** | **27.3** | **55.7** |

Table 7: Ablation on different textual encoders.

| Dataset | UCF-101 | | HMDB | | RareAct | |
|---|---|---|---|---|---|---|
| | Top-1 | Top-5 | Top-1 | Top-5 | Top-1 | Top-5 |
| Word2Vec | **40.1** | **66.3** | **27.3** | **55.7** | **11.5** | **32.8** |
| Sent2Vec | 39.7 | 65.4 | 23.9 | 51.2 | 7.3 | 22.7 |
| BERT | 31.8 | 58.9 | 22.2 | 45.6 | 8.3 | 22.9 |

## 4.4 Ablations

We perform several ablations on the UCF-101 and HMDB datasets to evaluate various components of our network.

**Text Refining Module** We evaluate the importance fine-tuning the textual features during training. We first remove the text refining module, and merge the semantic embedding, $\psi(y)$, with the visual features. This results in a Top-1 accuracy of 32.0% and 18.9% on UCF-101 and HMDB, respectively. Next, we use a single dense layer of size 1024 before merging features;his improves performance by 4.3% on UCF-101 and 5.7% on HMDB. Finally, we use our full 3-layer refining module which outputs a new textual encoding. We find that this leads to the best results with accuracies of 40.1% on UCF-101 and 27.3% on HMDB. This suggests that a single linear transformation (i.e. single dense layer) is insufficient to learn the visual semantics for given action classes, and a deeper refining module is required.

**Modality Merging** To determine the most effective technique for combining video and textual features, we use three common methods: multiplication, addition, and concatenation. For these experiments, all other components of the network remain unchanged while varying the merging method. We find that addition and concatenation perform similarly, but multiplication substantially outperforms them with a 5% improvement on UCF-101 and 3.3% improvement on HMDB.

**Importance of the Textual Encoder** Traditionally, ZSAR methods are trained using a Word2Vec encoder with max-pooling over the words of an action name. Pooling, however, can lead to loss of vital information. Newer text encoding methods like Sent2Vec or BERT [30] do not pool feature representation, but rather generate a single feature vector when given a sequence of words. We report scores on the UCF-101, HMDB, and RareAct datasets for various text encoders in Table 7. We find that for these datasets, Word2Vec tends to outperform these sentence based encoders; this can be attributed to the fact that the class names used in these datasets consist of a few words (the majority of classes are fewer than 2 words) so the max-pooling operation used with Word2Vec has minimal negative effect. This is supported by our experiments on the MEVA dataset, where Sent2Vec outperforms Word2Vec+Pooling since action classes are described with natural language descriptions.

## 4.5 Analysis

**Improved Textual Embeddings** We first analyse the text representations generated by our Text Refining module. We present a t-SNE [45] visualization of the Word2Vec encodings and the refined encodings in Figure 3. We find that the representations are changed based on the visual semantics of the given action class. For example, the original Word2Vec embedding for the action "Tennis Swing" is closer to the action "Swing" (i.e. child on a swing) than the action "Table Tennis Shot" which is visually more similar. Our Text Refining module, however, learns to bring "Tennis Swing" and "Table Tennis Shot" closer together in the feature space.

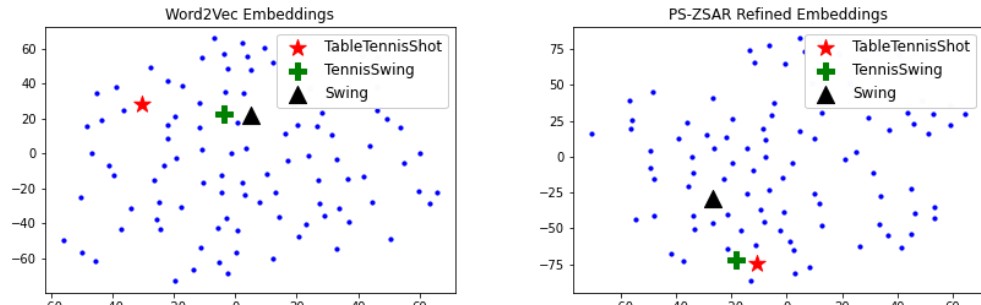

Figure 3: A comparison of the texual embeddings of UCF-101 classes of the previous method [4] and PS-ZAR, reduced to 2-dimensions via t-SNE. "Swing" and "Tennis Swing" both contain the word "Swing", and therefore averaging of Word2Vec embeddings falsely associates the two, while the more visually similar "Table Tennis Shot" class is further away. In PS-ZAR, the more visually similar classes are grouped closer together in the embedding space.

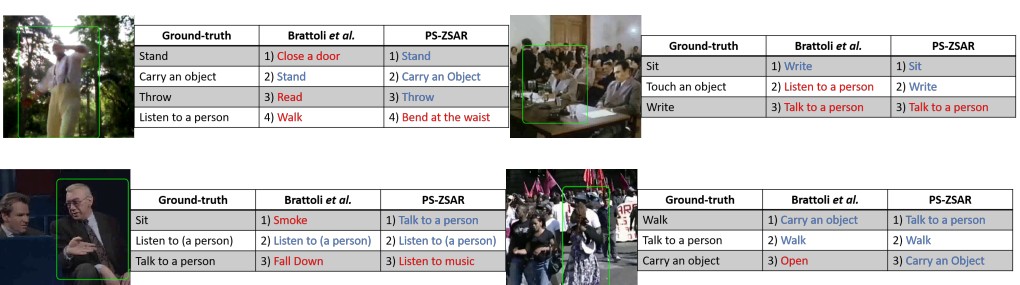

Figure 4: Qualitative results on multi-label samples from the AVA dataset for PS-ZSAR and the state-of-the-art baseline [4]. Our proposed approach is able predict semantically distinct classes, which leads to improved multi-label accuracy.

**Predicting Multiple Actions**   We argue that previous nearest-neighbor based ZSAR approaches are ineffective when dealing with multi-label data, since semantically distinct actions cannot be predicted together. We show some qualitative results to this effect in Figure 4 - for a given video we compare the top-5 predictions for the method [4] and our approach. We observe that our method successfully predicts semantically dissimilar classes while the previous approach can only predict classes which are similar.

# 5   Conclusion

In this work, we reformulate zero-shot action recognition such that it does not rely on nearest neighbor classification, but rather consists of a pairwise scoring function. Given a video and a set of action classes, our method predicts a set of probabilities for each class, allowing for semantically distinct classes to be predicted with high confidence. The proposed method improves on previous state-of-the-art zero-shot action detection models. We explore the use of feature merging as opposed to simply learning on a static textual embedding space, as well as multiple textual encoding schemes that preserve semantic relationships. Our results highlight the need for zero-shot learning models to learn both visual and textual spaces to properly represent unseen classes. The PS-ZSAR method achieves state-of-the-art performance on 5 zero-shot action classification benchmarks including in the challenging multi-label dataset AVA. Finally, we demonstrate our approach's ability to deal with real-world surprise activity detection by providing results on the MEVA dataset.

**Acknowledgments** This research is based upon work supported by the Office of the Director of National Intelligence(ODNI), Intelligence Advanced Research Projects Activity (IARPA), via IARPA R&D Contract No.  D17PC00345.  The views and conclusions contained herein are those of the authors and should not be interpreted as necessarily representing the official policies or endorsements, either expressed or implied, of the ODNI, IARPA, or the U.S. Government. The U.S. Government is authorized to reproduce and distribute reprints for Governmental purposes not withstanding any copyright annotation thereon.

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
