# Reformulating Zero-shot Action Recognition for Multi-label Actions (Supplementary Material)

**David S. Hippocampus**[*]
Department of Computer Science
Cranberry-Lemon University
Pittsburgh, PA 15213
`hippo@cs.cranberry-lemon.edu`

## 1 AVA Dataset Evaluation

### 1.1 Extracting Video Clips

Since the AVA dataset consists of multiple actors within one video and ZSAR focuses only on the classification task, we extract clips centered on the ground-truth bounding boxes for each actor in the video. Standard video models expect frame dimensions with the same height and width, so we crop a square region around the actor and resize it to the network specific dimensions ($112 \times 112$). We present some examples of AVA video frames with their annotations as well as the generated crops in Figure 1. This square crop can cause multiple actors to appear within one clip, as seen in the second example, but it ensures the aspect ratio of the person is not altered, which is necessary as this is the manner in which the video model is trained.

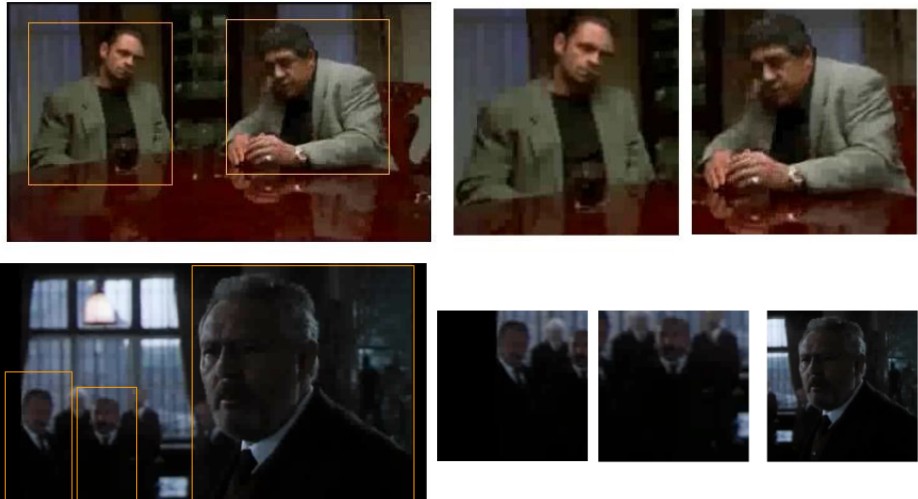

Figure 1: Example of original ground-truth bounding boxes (left) in the AVA dataset, with the cropped actors on the right.

---

[*]Use footnote for providing further information about author (webpage, alternative address)—*not* for acknowledging funding agencies.

35th Conference on Neural Information Processing Systems (NeurIPS 2021).

### 1.2 Generating Multiple Predictions and Confidences

As previous methods for ZSAR tend to be designed for single-label action classification, we adjust these methods to generate multiple predictions along with prediction confidences. For PS-ZSAR prediction confidences are obtained from the softmax probabilities output by our pair-wise similarity function. To obtain confidence scores from the method in Brattoli *et al.* [4], we apply a softmax operation on the inverse cosine distances between the video model's output and the semantic embeddings:

$$P(y|x) = \frac{\exp\left(-d\left(f_\theta\left(x\right), \psi\left(y\right)\right)/\gamma\right)}{\sum\limits_{y' \in \mathcal{U}} \exp\left(-d\left(f_\theta\left(x\right), \psi\left(y'\right)\right)/\gamma\right)}, \tag{1}$$

where $d$ is the cosine distance. As the distances between embeddings tend to be small, we use a temperature parameter $\gamma \leq 1$ increase distances before being passed through the softmax. We find that selecting $\gamma = 0.1$ leads to best results.

To obtain multiple predictions from a given method there are several approaches. One trivial approach is to select the top-k predictions for a given sample. The main issue with this approach is that it may over-predict classes when k is too large or under-predict when k is too small. Another approach is to predict all classes, in which case the mAP evaluation would ignore most low-confidence predictions. This alleviates the issue of under-predicting, but will always over-predict. Finally, we can predict classes based on a confidence threshold, in the manner described in equation 3 of the main paper.

Table 1: mAP Results on AVA Dataset

|  | Top-1 | Top-3 | Top-5 | No threshold | Threshold |
|---|---|---|---|---|---|
| Brattoli *et al.* [4] ($\gamma = 1$) | 1.6 | 2.1 | 2.4 | 3.1 | 6.2 |
| Brattoli *et al.* [4] ($\gamma = 0.1$) | 1.6 | 2.1 | 2.3 | 3.3 | 6.4 |
| Ours (word2vec) | 1.6 | 3.0 | 3.4 | 6.4 | 6.5 |
| Ours (sent2vec) | 1.5 | 3.0 | 3.5 | 5.7 | 7.0 |

We present results for all approaches in Table 1. It shows that the use of thresholding on predicted probabilities leads to best results. Interestingly, only the top-1 predictions for both methods achieve similar performance, but when it is increased to top-5, the gap between mAP scores increases. This poor performance is due to the nearest neighbor classification which does not allow semantically dissimilar classes to be predicted confidently. On the other hand, our approach can have multiple dissimilar classes in the top-5 predictions.

## 2 RareAct Evaluation

RareAct is a dataset compiled from rarely co-occurring nouns and verbs such as "microwave show" or "blend phone". It is meant to be "an evaluation dataset notably meant to be used to evaluate models trained on the HowTo100M dataset" [38]. We use RareAct in our work to evaluate how well zero-shot methods can deal with action classes which are extremely different from those seen during training. In the RareAct work [38], the authors propose different metrics (mWAP and mSAP). However, we evaluate our method using the top-1 and top-5 accuracy since the purpose of this work is to create a strong zero-shot classifier rather than learn a joint visual-textual model from a large-scale instructional dataset (i.e. HowTo100M).

## 3 Evaluation on UCF-101 and HMDB datasets using Random seeds

In Bratolli *et al.* [4], one of the primary methods of evaluation involves generating 10 different testing sets from UCF101 and HMDB by randomly choosing half of the classes. This is the standard evaluation in all works prior to [2], since lack of access to the Kinetics-700 dataset made testing on the full UCF-101 or HMDB dataset infeasible. However, we find that this metric is problematic as the results are dependant almost entirely on the random seed (implemented with numpy's rand package) used to choose which classes to test with. To illustrate this issue, we use 10k random seeds, and report the results in Table 2. The results for Brattoli *et al.* are obtained from the publicly available code and

| Dataset | UCF101 Class | MEVA Class |
|---|---|---|
| Class Name | BaseballPitch | person_opens_car_door |
| Encoder Input | "Baseball" "Pitch" | "A person opening the door to a vehicle. The only necessary track in this event is the vehicle. The vehicle door is not independently annotated from the vehicle. This event often overlaps with entering/exiting; however, can be independent or absent from these events." |
| Example | | Closing-00 |

Figure 2: **Example of the natural language descriptions of MEVA classes versus simple class names of UCF101 classes.** Note also for MEVA videos are captured through surveillance camera, and thus actions are lower resolution, as well as less visually apparent.

model weights. When results are averaged over all 10k seeds PS-ZSAR outperforms Brattoli *et al.*. Furthermore, we find that our method achieves higher accuracy on 58.3% of the seeds on UCF-101 and 76.5% of the seeds on HMDB .

Table 2: Evaluation on 50% of the UCF-101 and HMDB classes over 10k random seeds. Reported are the mean and standard deviation ($\mu \pm \sigma$).

| | UCF101 | HMDB |
|---|---|---|
| Brattoli *et al.*[4] | $39.3 \pm 4.3$ | $25.1 \pm 4.4$ |
| PS-ZSAR (ours) | $40.1 \pm 3.8$ | $27.3 \pm 4.0$ |

As our results show, Bratolli *et al.* [4] scores an average of 39.3 on UCF101 and 25.1 on HMDB. However, their reported results are 48.0 and 32.7 respectively, nearly two standard deviations above the mean. In the interest of reporting the most comparable results despite the drawbacks of this evaluation method, we searched for a seed that resulted in their method achieving as close to their reported scores as possible. In the main paper, we then reported our accuracy on that same seed: 49.2 and 33.8 for UCF-101 and HMDB respectively. As this evaluation protocol (i.e. selecting only 10 splits with 50% of the classes) can lead to noisy results, we argue future ZSAR should be evaluate on the entirety of UCF-101 and HMDB.

## 4 MEVA Dataset Activity Descriptions

Contrary to conventional video datasets which use class names to generate semantic embeddings, the MEVA dataset contain natural language descriptions of the action classes. For example, the action *carrying* has the description "A person carrying an object up to half the size of the person, where the person's gait has not been substantially modified. The object may be carried in either hand, with both hands, or on one's back" and the action *falling* has the description "A person falling by either (1) losing one's balance and possibly collapsing, or (2) moving downward from a higher to a lower level." These lengthy descriptions allow the ZSAR method to learn a richer semantic embedding which is useful for classifying surprise activities.

# 5   Method Limitations

We analyse how PS-ZSAR performs on the UCF-101 dataset to understand the limitations of the approach. We find that ZSAR methods achieve strong performance on certain classes, while many classes tend to be ignored and not predicted. We present 10 classes on which our method achieves 0% in Table 3. PS-ZSAR tends to predict classes which are visually similar to the target class. For instance, videos with the "Jump Rope" and "Jumping Jack" actions tend to be predicted as "Handstand Pushups" since all three actions involve similar motions (i.e. repetitive up and down motions). This is a limitation for not only our approach, but most ZSAR approaches. For example, Bratolli *et al.* [4] achieve 0% accuracy on 36 classes and PS-ZSAR achieves 0% accuracy on 22 classes. We believe solving this problem would be an interesting avenue for future work.

Table 3: Ten classes which PS-ZSAR performs worst on in the UCF-101 dataset. We include the class name, the accuracy, and the class predicted for most videos of the given class.

| Class Name | Most Predicted |
| --- | --- |
| Jump Rope | Handstand Pushups |
| Jumping Jack | Handstand Pushups |
| Hula Hoop | Tai Chi |
| YoYo | SalsaSpin |
| Front Crawl | Breast Stroke |
| Bowling | Basketball |
| Parallel Bars | Trampoline Jumping |
| Playing Daf | Head Massage |
| Playing Violin | Playing Flute |
| Pole Vault | Trampoline Jumping |