# OpenReview forum: "Reformulating Zero-shot Action Recognition for Multi-label Actions"
_NeurIPS.cc/2021/Conference — NeurIPS 2021 Poster_

### Official Review · Reviewer_NxRY · 2021-06-30

**Rating:** 5
**Confidence:** 5

**Summary:**

This work investigates zero-shot action recognition from a multi-label perspective. Rather than a nearest neighbour action selection, this work opts for a binary cross-entropy optimization. Experiments on common datasets as well as AVA and MEVA show the effectiveness of the approach.

**Limitations And Societal Impact:**

The authors mention that limitations are discussed, but this can not directly be found back in the paper. A more explicit discussion is beneficial. Societal impact is not listed and discussed explicitly, which could be relevant due to the topic of action recognition.

**Main Review:**

Strengths:

In zero-shot action recognition, the commonly used datasets are multi-class, with one action relevant per video. This work goes against this common assumption. Zero-shot multi-label recognition has been investigated multiple times before in the image domain, but hardly in the video domain. It is good to see additional effort in this direction.

Moreover, the approach is evaluated on five datasets, both multi-class and multi-label. The results are already good in the multi-class setting, which actually warrants further discussion as the approach is geared towards the multi-label setting. The main components are also studied individually to get a better sense of the most important elements in the overall architecture.

Weaknesses:

The paper is low on novelty. While multi-label zero-shot action recognition has not received a lot of attention, it has been investigated before as indicated by the authors. For the method itself, the notion of binary pairing at the video-level, instead of the class level is interesting, the rest of the approach follows standard binary cross-entropy optimization. The experiments and especially the evaluations on AVA and MEVA are new.

The method itself is not tied to actions. What makes the method interesting for actions? And how does it perform on other multi-label zero-shot problems? More importantly, how does the method compare to multi-label approaches from the image domain on actions? I.e. what happens when the approaches of e.g. [25-27] are used on the same (pre-)training data? Unfortunately, evaluations such as on AVA are lacking these baselines. It is therefore not possible to assess the effectiveness of the proposed approach.

The paper also misses a number of relevant zero-shot action papers for discussion and comparison. The paper focuses mostly on Zhu et al. [2] and Brattoli et al. [4] in the discussion and as baselines, but a number of relevant papers are missed in the process, e.g.:

[a] Mishra, Ashish, et al. "A generative approach to zero-shot and few-shot action recognition." 2018 IEEE Winter Conference on Applications of Computer Vision (WACV). IEEE, 2018.

[b] Jain, Mihir, et al. "Objects2action: Classifying and localizing actions without any video example." Proceedings of the IEEE international conference on computer vision. 2015.

[c] Tian, Yi, et al. "Aligned Dynamic-Preserving Embedding for Zero-Shot Action Recognition." IEEE Transactions on Circuits and Systems for Video Technology 30.6 (2019): 1597-1612.

[d] Mettes, Pascal, William Thong, and Cees GM Snoek. "Object priors for classifying and localizing unseen actions." International Journal of Computer Vision (2021): 1-18.

Especially [b] and [d] are relevant as they do not transfer from seen to unseen actions, but from objects to unseen actions. A score is computed for each action separately. As these works focus on the multi-class setting, the argmax is taken, but it is equally possible to use a threshold as done in this paper. On UCF-101, [d] scores close to Brattoli et al., making it a viable candidate for comparison on AVA.

Conclusion:

This paper shins a brighter light on the multi-label recognition of actions in the zero-shot action, which is indeed needed. An approach that reasons on video-label pairs is proposed to make this possible. Experiments one five datasets show that the method is effective overall. The novelty of the paper is however low, as the task itself is not new, the method is not really new and especially not specific to actions, while the experimental comparisons are lacking. Moreover, relevant related work is not cited and discussed. Please address these issues in the rebuttal.

**Time Spent Reviewing:**

4

---

> ### Author Response · Authors · 2021-08-10
> **Reposne to Reviewer NxRY**
>
> We thank the reviewer for their time and thorough comments.
>
> ---
>
> **The paper is low on novelty. While multi-label zero-shot action recognition has not received a lot of attention, it has been investigated before as indicated by the authors. For the method itself, the notion of binary pairing at the video-level, instead of the class level is interesting, the rest of the approach follows standard binary cross-entropy optimization.**
>
> We believe that the simplicity of our proposed model allows it to be a compelling system for future works to expand upon. Our work reformulates the problem toward a learned pair-wise scoring network, which allows for fine-tuning of textual encoding. Previous nearest neighbor based approaches did not allow for this step, which lead to the weaknesses in action class representation. This is depicted in Figure 3, where more visually similar classes are  mapped more closely when fine-tuned. We think that moving away from the nearest neighbor paradigm opens up opportunities for many future improvements in both single-label and multi-label ZSAR.
>
> **The method itself is not tied to actions. What makes the method interesting for actions? And how does it perform on other multi-label zero-shot problems?**
>
> Although our approach is general and it can be applied to images with some modifications, our focus is in the video domain. We show that our approach performs well on several existing and new difficult zero-shot action recognition benchmarks in both single-label and multi-label settings. However, we can expand our related works to include additional image-based zero-shot approaches.
>
> **The paper also misses a number of relevant zero-shot action papers for discussion and comparison**
>
> We will include these papers in our related works. Works pre-trained on object recognition tasks are difficult to compare to ours, in terms of zero-shot action recognition, as many of these systems have seen objects that appear in the testing set. For example, UCF101 contains actions utilizing many different instruments and sports apparatuses that would likely appear in the ImageNet pretraining set. The class filtering procedure we, and previous works [4,43], follow would require these training examples be removed to be comparable.

---

> > ### Comment · Reviewer_NxRY · 2021-08-26
> > **Open questions**
> >
> > After the rebuttal, the following question remain unanswered:
> > * While the focus on videos specifically is fine, it is unclear whether the proposed approach works well compared to current image-based solutions.
> > * The answer about the comparison to object-based approaches is not satisfactory. There are no object labels in UCF101 so there is no need to filter actions and a comparison to object-based zero-shot approaches is feasible.
> >
> > To clarify: while the latter point is unlikely to work well since it is not designed for the multi-label setting, comparisons to multi-label approaches from the image domain would strengthen the paper.

---

> > > ### Author Response · Authors · 2021-08-30
> > > **Response to open questions**
> > >
> > > **While the focus on videos specifically is fine, it is unclear whether the proposed approach works well compared to current image-based solutions.**
> > >
> > > We agree that image based methods can be extended to action recognition in video domain, however, we only focus on videos in this work and have compared with most state-of-the-art methods in the zero-shot action recognition literature and we find that our approach performs well compared to these strong baselines.
> > >
> > > **The answer about the comparison to object-based approaches is not satisfactory. There are no object labels in UCF101 so there is no need to filter actions and a comparison to object-based zero-shot approaches is feasible.**
> > >
> > > Although there are no explicit object labels in UCF-101, the object names are often present within the action label. For example, "Horse Race" and Horse Riding" are class names in UCF-101, and the object "horse" is present in the MSCOCO dataset which is used to train the object-based zero-shot method [d]. Therefore, for a fair zero-shot comparison we would have to perform filtering operation on MSCOCO training set.

---

> > > > ### Comment · Reviewer_NxRY · 2021-09-01
> > > > **End of rebuttal**
> > > >
> > > > I thank the authors for engaging in the discussion. We have also discussed the paper internally and based on that discussion, I will move to a 5. I am a bit disappointed in the unwillingness to compare to relevant baselines that have previously been investigated in the image domain. The action baselines are multi-class predominantly and not multi-label. The existing multi-label zero-shot comparisons would provide insights in the effectiveness of the proposed approach and I would urge the authors to include a few in the paper.
> > > >
> > > > As stated earlier, the object-based baselines are less relevant since they are not multi-label either and will not directly affect the score. While this is neither the time or place for such a discussion, I do not agree with the explanation for excluding object-based baselines. Recognizing an object does not provide a direct link to an action. The listed example of a horse can still lead to many different actions. See e.g. [b] for a work on object-based zero-shot action recogniton on the entire UCF-101 dataset. Similarly, you would still use a color-based attributes in the image domain even in the presence of e.g. a brown bear category. I find the proposed plan of the authors to include the references sufficient however.

---

### Official Review · Reviewer_CrDZ · 2021-07-15

**Rating:** 6
**Confidence:** 4

**Summary:**

This work has two main contributions: (1) reformulation of the zero shot action recognition task for the multi-label case, and (2) a new model motivated by the authors idea that nearest neighbor is sub-optimal on a static semantic space. A pairwise scoring function is proposed in this work which first maps both visual and semantic embeddings into a joint space to predict match/no match probabilities of a 3D-CNN based visual representation and a Word2Vec/Sent2Vec based text representation. Their approach is evaluated not only on multi-label action recognition (where the zero shot performance is evaluated for the first time) dataset but also on single-label action recognition dataset, achieving state-of-the-art results. Evaluation protocols for single-label zero-shot action recognition datasets, UCF-101, HMDB, RareAct is proposed and another two evaluation protocols for multi-label zero-shot action recognition based on MEVA and EVA datasets are proposed in this work.

**Limitations And Societal Impact:**

Since this work focuses on zero-shot recognition it makes a good step towards solving the data efficiency problem of the deep CNNs. The models are evaluated on public datasets and we do not see any ethical concerns beyond the conventional issues linked to visual recognition.

**Main Review:**

Several highlights and shortcomings are found during review.
Highlights:
1)	The authors reformulate zero-shot action recognition for a multi-label setting, which in my opinion is an important task.
2)	Interesting although a relatively simple idea for a new approach:  learning to infer semantic similarities given text label and a visual representation as input. The authors argued that the most leveraged nearest neighbor-based approach in zero-shot recognition is sub-optimal and proposed a scoring function-based approach to achieve one-stage zero-shot multi-label prediction.
3)	For the dataset contributions, one evaluation protocol based on UCF-101 and HMDB leveraged random seed, and another one based on full RareAct dataset for single-label zero-shot action recognition are given. Besides these, two evaluation protocols for multi-label zero-shot recognition based on EVA and MEVA datasets are also given in this work.
4)	Extensive evaluation results: Ablations for different refining modules of text encoder, such as no refining, single layer refining and full model refining, for text encoder and video encoder feature fusion modalities, such as concatenation, addition and multiplication and for different textual encoders such as Word2Vec, Sent2Vec and BERT are illustrated in their work, indicating clearly which modality setting is better for label text feature extraction and fusion between video embedding and text embedding.
5)	State-of-the-art results are achieved for both: the newly proposed multi-label and the existing single-label zero shot action recognition.
Shortcomings:
1)	Since the authors argue that for semantic similarity nearest neighbor approach is sub-optimal, for single-label and multi-label zero-shot recognition there is no comparison experiments between nearest neighbor-based approach and score function-based approach proposed by authors leveraging both semantic representation and video representation.
2)	The parameterized function which maps an input video to the semantic space illustrated in line 109 has the same representation with the scoring function mentioned in line 115, which may cause confusion.
3)	There is a small wrong typing in line 248.
Question:
1)	There is a question for formular (5) under line 142, does N forward passes be leveraged or only one forward pass for confident score prediction of different classes? Do the authors reinitialize the fc layer for different classes’ confidence score prediction?
After the analysis among highlights, shortcomings and question, this is a good work, which provided interesting multi-label and single-label zero-shot recognition evaluation protocols on the well-known datasets and proposed a score function-based approach to take place with the common leveraged nearest neighbor method for zero-shot action recognition considering the similarity between video embedding and semantic embedding, while there are still several problems found in this work. The efficacy verification of their score function has not been illustrated in experiments by comparing the zero-shot recognition performance between nearest neighbor and score function proposed in their work while using same feature embedding setting. And also the forward pass of the score function is not illustrated clearly as the aforementioned question part. According to the above several analyses, this paper is recommend scored as marginal accept and we would kindly ask for clarification on the posed question.


We have read the other reviews and the author rebuttal and will stay with our initial assessment.


**Time Spent Reviewing:**

ca five hours

---

> ### Author Response · Authors · 2021-08-10
> **Respose to Reviewer CrDZ**
>
> We would like to thank the reviewer for taking the time to review our work as well as their constructive feedback.
>
> ---
>
> **6: Since the authors argue that for semantic similarity nearest neighbor approach is sub-optimal, for single-label and multi-label zero-shot recognition there is no comparison experiments between nearest neighbor-based approach and score function-based approach proposed by authors leveraging both semantic representation and video representation.**
>
> Brattoli et al. [4] is a nearest neighbor based approach that uses the same R(2+1)D-18 video backbone and same text features (w2v embeddings with average pooling) as our experiments. Therefore, it is a strong nearest neighbor baseline, which is present in row 2 of Table 1 for the single-label case an and row 1 of Table 3 for the multi-label case.
>
> **7, 8:Confusing notation and Typo**
>
> Thank you for pointing this out, we will rename the function to alleviate confusion and fix the typo.
>
> **9: Does N forward passes be leveraged or only one forward pass for confident score prediction of different classes?**
>
> In our implementation, we do not require N forward passes. Instead, due to broadcasting in PyTorch, we can perform all computations as a single forward pass with a batch of different classes. For example, given N videos and C action classes, the network can output confidence scores of shape (N, C) in a single forward pass. Also, we do not reinitialize the fully connected layer for different classes’ confidence score prediction.

---

### Official Review · Reviewer_KU7P · 2021-07-20

**Rating:** 6
**Confidence:** 5

**Summary:**

This paper proposes a strategy for zero-shot action recognition. One of the contribution is to design the model such as it directly outputs a score given an unseen class name. This is different from past work that instead rely on nearest neighbor in a joint text-video space. The authors explore various ways to combine the visual and semantic class name in the architecture. They report results on multiple datasets UCF-101, HMDB, RareAct, AVA and MEVA where they show improvements over previous work trained on the same dataset [4].

**Ethical Concerns:**

None.

**Limitations And Societal Impact:**

Yes.

**Main Review:**

## Strengths

- Zero-shot action classification is an important problem to tackle
- The paper is clearly written
- Compelling numbers on HMDB51 and UCF101 compared to previous work using the same setting
- The idea of outputting a score directly for a given class allows to explore more complex interactions between the visual signal and the text contained in the class name than using simple nearest neighbors in a vector space. In particular, according to Table 6, it seems that the multiplicative interaction is the important architecture design of the paper.

## Weaknesses and possible improvements

- **More complex interactions**: Right now only concatenation, sum and multiplicative interactions are explored. As it seems to be quite important to improve the performance, have you explored other architecture for interactions? One could for example use attention mechanisms to combine the visual and the text features (e.g. similar to what is used in works such as [a] VilBERT).

[a] ViLBERT: Pretraining Task-Agnostic Visiolinguistic Representations for Vision-and-Language Tasks, Lu et al.

- **What happens on the regular classification accuracy?** Even if not the main focus of the work, I believe it would be worthwhile to provide the accuracy numbers obtained on Kinetics700 when training with that technique (the one illustrated in Figure 2) compared to standard softmax loss. Can we expect improved numbers on standard action recognition as well thanks to the information contained in word2vec and the proposed interaction architecture? Even if the result is not compelling I believe that would be an interesting datapoint to obtain.

- **Comparing/discussing recent zero-shot work**: I think it would be worth discussing a recent trend in zero-shot recognition which consists in training on large corpora of webly supervised text+image/video datasets (see [b,c,d] below) and discuss how the two methods could potentially be merged together. In particular even if not comparable because the training datasets are very different, [c] obtains 80.3 on UCF101 zero shot even though it was only trained on still images (compared to 40.1 in the paper). Similarly [b] was trained on noisy instructional videos (HowTo100M dataset) and still perform quite well on the RareAct dataset (see next point).

- **RareAct correct evaluation metric**: Please provide the metric of RareAct used in the paper (wAP). The authors provide an implementation of this score in [here](https://github.com/antoine77340/RareAct/blob/master/compute_score.py) and this should be compatible with your model as you only need to provide a matrix score containing the score for each video and for all classes. That way you will be able to compare to [b] and [c] which would give a point of reference in the future.

- [4] further improves its results using SUN augmentation. Have you considered using this as well in your case to verify the improvements still hold?

- Missing related work:
  - On the multiplicative interactions, there are a few work that would be worth mentioning:
    - FiLM: Visual Reasoning with a General Conditioning Layer, Perez et al.
    - Squeeze-and-Excitation Networks, Hu et al.
    - Learnable pooling with context gating for video classification, Miech et al.
  - Some recent work that can perform zero-shot (mentioned above) and that would be worth discussing:
    - [b] End-to-end learning of visual representations from uncurated instructional videos, CVPR2020 (for comparison on RareAct)
    - [c] Learning Transferable Visual Models From Natural Language Supervision, Radford et al. (provide comparison on UCF101 and RareAct).
    - [d] Scaling Up Visual and Vision-Language Representation Learning With Noisy Text Supervision, Jia et al (ICML21).

## Overall assessment

The paper proposes an interesting idea and the results seem to suggest that the proposed changes are effective as it improves over a recent baseline [4]. For this reason I am leaning towards acceptance however there are a few things that could improve the paper further and I will wait on the rebuttal to make my final decision.


**Time Spent Reviewing:**

4

---

> ### Author Response · Authors · 2021-08-10
> **Response to Reviewer KU7P**
>
> We thank this reviewer for their positive feedback as well as their time. We address the various comments below.
>
> ---
>
> **More complex interactions: Right now only concatenation, sum and multiplicative interactions are explored. As it seems to be quite important to improve the performance, have you explored other architecture for interactions? One could for example use attention mechanisms to combine the visual and the text features (e.g. similar to what is used in works such as [a] VilBERT).**
>
> As this work mainly deals with reformulating ZSAR to not rely on nearest neighbor classification we do not focus on optimizing performance on the merging operation. We agree that more complex approaches for combining text and video features (e.g. context gating [e], feature-wise linear modulation [f], or attention-based methods [a]) may improve performance, but in our experiments we mainly explore concatenation, summation, and multiplication as methods for merging features.
>
>
> **What happens on the regular classification accuracy? Even if not the main focus of the work, I believe it would be worthwhile to provide the accuracy numbers obtained on Kinetics700 when training with that technique (the one illustrated in Figure 2) compared to standard softmax loss.**
>
> As we train our network on all videos from a subset of the Kinetics-700 (662 action classes that do not overlap with the test set classes), we cannot directly evaluate on the standard Kinetics-700 test set. Currently, we are running an experiment with our model to evaluate the supervised classification accuracy (training and evaluating on the conventional train/test splits of Kinetics-700). Once this evaluation is done, we can include this in the supplemental materials.
>
>
> **Comparing/discussing recent zero-shot work: I think it would be worth discussing a recent trend in zero-shot recognition which consists in training on large corpora of webly supervised text+image/video datasets (see [b,c,d] below) and discuss how the two methods could potentially be merged together.**
>
> These zero-shot works learn from large amounts of data consisting of video-text or image-text pairs and they show strong performance on downstream datasets, like UCF-101 and RareAct. In this work, however, we follow the zero-shot action recognition paradigm outlined in [43]  such that there is no overlap of action classes in training and test sets. Since these works [b,c,d] are trained on an uncurated data, it is likely that many of the text sequences are similar to the action classes present in the downstream dataset. For example, HowTo100M contains many "Sports and Fitness" videos, of which many could overlap with UCF-101 or HMDB (e.g. Basketball and Bowling videos are present in HowTo100M and UCF-101). Therefore, comparisons between our approach and these works would be invalid. We think that discussing these differences between zero-shot methods in the related works would be useful, and we will include it in the final version.
>
> **Please provide the metric of RareAct used in the paper (wAP). The authors provide an implementation of this score in here and this should be compatible with your model as you only need to provide a matrix score containing the score for each video and for all classes**
>
> Following the reviewer's suggestion, we have evaluated on RareAct using the wAP metric. Our method achieves mWAP of 5.9\%. This score, however, is not comparable to [b,c] as both of these works are trained on a larger datasets (like HowTo100M) with greater variation in terms of visual and textual data. For example, "chocolate" and "oven" concepts are not present in Kinetics-700, but are abundant in HowTo100M cooking videos. We can include this score in the camera ready work, so that future ZSAR methods trained on Kinetics can have a point of reference.
>
> **That way you will be able to compare to [b] and [c] which would give a point of reference in the future.[4] further improves its results using SUN augmentation. Have you considered using this as well in your case to verify the improvements still hold?**
>
> In our experiments on only Kinetics, we show that our method outperforms [4] even when their network has additional SUN pretraining (40.1\% top-1 accuracy vs. 39.8\% in [4]). As the improvement gains were shown to be relatively small (about 2\% improvement in top-1 accuracy was found in Brattoli et al. [4]), we did not perform pretraining on SUN images. However, in response to this comment, we are running pretraining on SUN and will include the results in the camera ready version.
>
> [a] ViLBERT: Pretraining Task-Agnostic Visiolinguistic Representations for Vision-and-Language Tasks, Lu et al.
>
> [b] End-to-end learning of visual representations from uncurated instructional videos, CVPR2020 (for comparison on RareAct)
>
> [c] Learning Transferable Visual Models From Natural Language Supervision, Radford et al. (provide comparison on UCF101 and RareAct).
>
> [d] Scaling Up Visual and Vision-Language Representation Learning With Noisy Text Supervision, Jia et al (ICML21).
>
> [e] Learnable pooling with context gating for video classification, Miech et al.
>
> [f] FiLM: Visual Reasoning with a General Conditioning Layer, Perez et al.

---

> > ### Comment · Reviewer_KU7P · 2021-08-26
> > **Thank you for your responses**
> >
> > > **Once this evaluation is done, we can include this in the supplemental materials.**
> >
> > Any chance you already got these numbers and could share them here? Thanks!
> >
> > > **However, in response to this comment, we are running pretraining on SUN and will include the results in the camera ready version.**
> >
> > Same question here, if you have these results already it would be great to share them here.

---

> > > ### Author Response · Authors · 2021-08-28
> > > **Additional Results**
> > >
> > > **What happens on the regular classification accuracy?**
> > >
> > > We have trained our model in the fully-supervised setting on all classes of Kinetics. With our method we achieve a clip top-1 accuracy of 55.5% on the validation set. This is similar to the reported Clip@1 accuracy (56.8%) reported in Table 1 of [31]. However, these two scores are not one-to-one as the training setups have differences (e.g. data preprocessing, batch size, learning rate/scheduler, etc.). We are currently in the process of training the standard R(2+1)D classifier using the same training setup as our model, so that we can have a fairer baseline comparison.
> > >
> > >
> > > **Pre-training with augmented image dataset (SUN)**
> > >
> > > We have conducted additional experiments following the procedure in [4] where the model is first pre-trained on the augmented SUN images and then fully trained on the Kinetics dataset. Note, we perform the same class filtering procedure on the SUN dataset. Below are our results on both UCF-101 and HMDB. From these results, we see that the improvement holds when compared to Bratolli et al. [4]. There is a consistent 2-3% improvement when the model is pre-trained with the SUN data. These results will be included in the final camera-ready version.
> > >
> > >
> > > | UCF-101      | Top-1 | Top-5 |
> > > |:---------------|:----:|:----:|
> > > | Brattoli et al. [4] (Kinetics) | 37.6 | 62.5 |
> > > | Brattoli et al. [4] (SUN+Kinetics) | 39.8 | 65.6 |
> > > | Ours (Kinetics) | 40.1 | 66.3 |
> > > | Ours (SUN+Kinetics) | 43.2 | 69.5 |
> > >
> > >
> > > | HMDB      | Top-1 | Top-5 |
> > > |:---------------|:----:|:----:|
> > > | Brattoli et al. [4] (Kinetics) | 26.9 | 49.8 |
> > > | Brattoli et al. [4] (SUN+Kinetics)  | - | - |
> > > | Ours (Kinetics) | 27.3 | 55.7 |
> > > | Ours (SUN+Kinetics) | 29.5 | 58.1 |

---

### Official Review · Reviewer_3jdc · 2021-07-25

**Rating:** 6
**Confidence:** 4

**Summary:**

In this work, authors reformulate zero-shot action recognition such that does not rely on nearest neighbor
classification, but rather consists of a pairwise scoring function. Given a video and a set of action
classes, the method predicts a set of probabilities for each class, allowing for semantically distinct
classes to be predicted with high confidence. The proposed method improves on previous state-
of-the-art zero-shot action detection models and is tested on UCF101, HMDB, AVA datasets among others.

**Ethical Concerns:**

None, if any they would be about making sure word embedding models are trained properly without any biases.

**Limitations And Societal Impact:**

Limitations - The main limitation of the proposed work is it is a simple extension of backpropogating through two different models instead of freezing one model's output (text embedding) and using it with a cross entropy loss.  The video encoding, text encoding, jointly training the models and the final cross entropy loss are all very well studied modules and simpler extensions of existing works. So, the technical novelty of the proposed approach is a bit limited even though the results are good. The simplicity of the method though could be a positive in-spite of these limitations.

On Societal impact - N/A. Authors didn't touch on it in anyway but the subject area and the models built seem fairly safe from this angle that authors didn't have to think deeply about the societal impact.

**Main Review:**

The proposed method is tackling the problem of Zero shot action recognition in videos. The core idea is to learn both video representations and semantic text embeddings in and end to end manner with and merge the representations with a cross entropy loss to handle multi label prediction. The authors did a good job describing the literature, writing a clean paper (a bunch of places with grammar issues but not major), doing significant experimental analysis of the method on various small and large scale datasets and also have ablation studies to backup the utility of the proposed method.

The only limitations if any would be that the proposed method is a simple extension of existing methods and for NeurIPS standards might not have the deep technical contributions. Nevertheless, i believe the work is valuable and is tackling an important problem of zero shot learning in videos.

**Time Spent Reviewing:**

1

---

> ### Author Response · Authors · 2021-08-10
> **Response to Reviewer 3jdc**
>
> We thank this reviewer for their positive feedback as well as their time. We address the comment about novelty below.
>
> ---
>
> **The only limitations if any would be that the proposed method is a simple extension of existing methods and for NeurIPS standards might not have the deep technical contributions.**
>
> The main contribution of our paper is the reformulation of zero-shot action recognition from a standard nearest-neighbor classifier to a pair-wise scoring function which produces class-level confidence scores directly. With respect to ZSAR, this allows our framework to move away from the standard approach of having one fixed modality (i.e. text encoding) to having both encodings learned during training. Our qualitative results (Figure 3) and experimental evaluations (Tables 3 and 4) show that fixed textual embeddings used in nearest neighbor approaches result in multiple false associations between text and video, leading to poor results when multiple labels are present within one sample.
>
> Although each of the components of our method (video encoding, text encoding, jointly training, and cross entropy) have been studied separately, we show that the combination of these modules in a ZSAR framework outperforms existing nearest-neighbor based approaches. We think that the simplicity of our proposed method allows it to be a compelling system that can be expanded upon. Whereas nearest neighbor approaches require textual encodings to be fixed and relegate most performance gains to the improvement of the video features, a pairwise scoring system opens up future works to more robust and sophisticated uses of text as a feature of the architecture. Our proposed method can be used as a foundation for ZSAR methods to improve not just the video encoder, but also the other components of our method (i.e. the refining module, modality merging, and the scoring function).

---

### Decision · Program_Chairs · 2021-09-27

**Decision:**

Accept (Poster)

**Comment:**

There was active engagement between the authors and reviewers following the rebuttal.

It is important that the authors update the paper with the new results requested by the reviewers, as well as update the related work section with (i) missing citations for video ZSL, and (ii) related work on image ZSL, that should also be discussed.

One other issue that could be more thoroughly discussed is the significance of the type of interaction, where multiplicative gives a subtantial performance improvement over the other choices.